# Efficiency of Four Extraction Methods to Assess the Bioavailability of Oxyfluorfen to Earthworms in Soil Amended with Fresh and Aged Biochar

Chi Wu [1,2], Lan Zhang [2], Liangang Mao [2], Lizhen Zhu [2], Yanning Zhang [2], Hongyun Jiang [2], Yongquan Zheng [1,*] and Xingang Liu [2,*]

1   Engineering Research Center for Environment-Friendly Agricultural Pest Management, College of Plant Health and Medicine, Qingdao Agricultural University, Qingdao 266109, China; chiwu2022@163.com
2   State Key Laboratory for Biology of Plant Disease and Insect Pests, Institute of Plant Protection, Chinese Academy of Agricultural Sciences, Beijing 100193, China; lanzhang@ippcaas.cn (L.Z.); lgmao@ippcaas.cn (L.M.); zhulizhen1004@163.com (L.Z.); zhangyanning@caas.cn (Y.Z.); hyjiang@ippcaas.cn (H.J.)
*   Correspondence: zhengyongquan@qau.edu.cn (Y.Z.); liuxingang@caas.cn (X.L.)

**Abstract:** Due to its high persistence in soil, oxyfluorfen has negative effects on environmental and human health. To reduce soil contamination and impacts on non-target organisms, biochar is introduced into soils to immobilize and sequestrate oxyfluorfen as a remediation practice. Three types of soils common in China were selected and biochar (rice hull, BCR) was added to investigate the desorption and bioavailability of oxyfluorfen after aging BCR for 0, 1, 3, and 6 months. Four chemical extraction methods were used to predict oxyfluorfen bioavailability. Results indicated that after addition of 0.5–2% unaged BCR, the desorption values of oxyfluorfen increased from 64–119 to 176–920 $(\mu g/g)/(mg/L)^n$ in the three soils compared with unamended soil. The bioaccumulation factor (BCF) values of oxyfluorfen in earthworms decreased from 0.80–1.7 to 0.10–1.56 after BCR addition. However, the desorption values decreased from 170–868 to 144–701 $(\mu g/g)/(mg/L)^n$ after aging. The bioavailability of oxyfluorfen in earthworms also increased after the aging treatments, while the BCF was still lower than with unaged BCR. The reduced BCF indicated lower exposure risk of oxyfluorfen to earthworms after amendment with biochar, even after aging 6 months. The bioavailability after extraction by Tenax showed a high linear correlation with uptake in earthworms, even after the 6-month aging treatment ($R^2 > 0.80$). Consequently, BCR could be a practical method to remediate contaminated soil and the 6h Tenax method could be a sensitive and feasible tool to assess the bioavailability of oxyfluorfen in soil.

**Keywords:** chemical extraction; oxyfluorfen; bioavailability; earthworm; aged biochar

## 1. Introduction

Extensive use of pesticides in agriculture has negative effects on the development of crops as well as non-target organisms [1,2]. Contamination by pesticides in soil poses severe environmental problems, such as pollution in food and water, which increase the risks to humans and biota [3–5]. As a diphenyl ether herbicide, oxyfluorfen is most widely used as an inhibitor of protoporphyrinogen oxidase (PPO) to control the annual broadleaf weeds and grasses in tropical and sub-tropical crops and is applied at pre- or post-emergence in agriculture [6–8]. However, oxyfluorfen is currently considered a highly toxic herbicide because of its toxicological effects on two fish species [9]. Furthermore, development of human erythroblastic progenitors (BFU-E/CFU-E: burst forming unit-erythroid and colony forming unit-erythroid) and hemoglobin synthesis showed a cytotoxic effect at $10^{-2}$ M of oxyfluorfen [10]. Oxyfluorfen also had negative impacts on pumpkins and showed phytotoxic damage in rice [11,12]. Furthermore, due to persistent use in paddy

fields, oxyfluorfen may contaminate soil [13] and groundwater as a result of drift and runoff [14,15]. Moreover, due to its medium-to-high soil persistence [13,16,17], pollution by oxyfluorfen poses potential risks to the safety of the soil environment. Therefore, it is urgent to reduce contamination by oxyfluorfen in soil environments.

Biochar has been used as a remediation agent to immobilize organic pollutants due to its high sorption capacity. Biochar also changes the physical properties of soil, and as a result, it can influence the environmental fate of organic pollutants in soil [18,19]. In a previous study, amendment with biochar reduced the bioavailability and toxicity of organic pollutants in sediment and soil due to its high porosity and large specific surface area [4,20,21]. However, the properties of biochar vary with time after its application in soil, depending on soil properties and environmental conditions, meaning that it may be regarded as a soil pollutant [22–25]. Thus, it is necessary to select the optimum biochar for different soil types even after an aging process. As Li et al., reported, bioaccumulation of acetochlor in plants increased when biochar aged for 20 days was used, causing high potential exposure risks to the environment [26]. Fomesafen also showed greater desorption, leaching, and bioavailability after biochar was aged for 6 months [27]. Therefore, it is necessary to find suitable biochar for practical remediation applications.

Bioavailability is the available part of a compound in the environment that could be taken up by an organism. Generally, concentrations of pollutants accumulate in plants, and earthworms could absorb the bioavailable part of organic pollutants. The bioavailability of contaminants is influenced by the physical and chemical properties of soil as well as the biological interactions associated with soil. In addition, because methods to extract pollutants from organisms are laborious, time-consuming, and expensive [28], finding practical methods to predict bioavailability is another challenge for risk assessment. Chemical extraction methods, such as mild-solvent extraction, solid-phase extraction, sorbent-assisted desorption, and supercritical fluid extraction methods, have been demonstrated as practical tools to estimate exposure concentrations for risk assessment [29–31]. The single-point Tenax extraction has a strong correlation with bioaccumulation in aquatic invertebrates and toxicity results [32–34]. However, previous reports also indicated that the bioavailability of polycyclic aromatic hydrocarbon (PAH) to earthworms was over-predicted by HPCD and Tenax 6h extraction [35]. Different chemical extraction methods result in different efficiencies of extraction [28]. Therefore, it is necessary to select a suitable chemical extraction method to predict bioavailability.

The aims of this study were to (1) comprehensively assess the risk of oxyfluorfen in three typical soils in China after amendment with biochar, based on desorption ability and bioaccumulation; (2) select a suitable chemical extraction method to evaluate the bioavailability; and (3) use the optimum chemical extraction method to assess bioavailability in the amended soil and soil after aging. For this purpose, a desorption experiment, a bioaccumulation experiment, and an assessment of bioavailability were conducted in three soils with biochar aged 0, 1, 3, and 6 months.

## 2. Materials and Methods

### 2.1. Materials and Chemicals

Oxyfluorfen (97%, analytical grade) was purchased from the China Reference Material Center (Beijing, China). High-performance liquid chromatography (HPLC)–grade N-hexane (Beihua Fine-chemicals Co., Ltd., (Beijing, China)) was used to prepare a stock solution (10,000 mg/L) of oxyfluorfen. Other reagents, such as sodium azide and calcium chloride (analytical grade), and HPLC-grade solvents were obtained from Beihua Fine-chemicals Co., Ltd., (Beijing, China). Earthworms *Eisenia foetida* (Savigny, 1826) were obtained and cultured at the Chinese Academy of Agricultural Sciences (Beijing, China).

### 2.2. Soil and Biochar with Aging Treatment

Biochar (BCR) made using rice hull and pyrolyzed at 500 °C was obtained from the Zhejiang Biochar Engineering Technology Research Center. Three types of soil samples

were collected from the Hebei (HBS), Hunan (HNS), and Heilongjiang (DBS) provinces in China. The collection areas had no application of oxyfluorfen in recent years. The depth of soil samples was 0 to 10 cm. The parameters of soil, including the location, classification, and physicochemical properties, are given in Table 1. The mixtures of soil and BCR were cultured at room temperature for 1, 3, and 6 months for the aging treatment. The moisture content of soils was maintained within 60% maximum water holding capacity (MWHC) during the aging process. An Inspect S50 scanning electron microscope was operated for scanning electron microscopy (SEM) under high vacuum (10 kV) using an Everhart–Thornley Detector.

**Table 1.** Physical and chemical properties of tested soils and biochar.

| | | | | | Physical and Chemical Properties of Tested Soils and Biochar | | | | | | |
|---|---|---|---|---|---|---|---|---|---|---|---|
| | | | | | OM | OC | TN | | Clay | Silt | Sand |
| Soil | Source | Location | pH | CEC (cmol/kg ( + )) | (%) | (%) | (mg/kg) | Texture | (%) | (%) | (%) |
| HNS | Hunan | N 28°19′, E 113°9′ | 4.85 | 24.8 | 1.45 | 0.84 | 574 | Loamy clay | 43.2 | 24.7 | 32.1 |
| HBS | Hebei | N 39°30′, E 116°36′ | 7.55 | 9.04 | 1.69 | 0.98 | 273 | Sandy loam | 14.5 | 12.3 | 73.2 |
| DBS | Heilongjiang | N 45°47′, E 126°29′ | 6.59 | 27.8 | 3.84 | 2.23 | 1740 | Clay loam | 21.9 | 23.1 | 55 |
| Biochar | Feedstock | pH | C (%) | O (%) | H (%) | N (%) | O/C (%) | H/C (%) | (O + N)/C (%) | Ash (%) | SSA (m$^2$/g) |
| BCR | Rice hull | 6.96 | 33.6 | 13.53 | 2.22 | 0.31 | 0.3 | 0.79 | 0.31 | 50.34 | 95.67 |

SSA: specific surface area determined by the BET adsorption method; PV: pore volume; PS: pore size. OM (organic matter) = OC (organic content)/1.724; TN: total nitrogen content.

### 2.3. Desorption Experiment of Oxyfluorfen

The desorption experiment of oxyfluorfen was conducted using the conventional technique [36]. Desorption was conducted immediately after the adsorption experiment described in our previous study [37]. A total of 1.1 g/L $CaCl_2$ was added and then the mixture was shaken for at least 36 h for equilibration of soil [38]. The soil suspensions were then centrifuged, and 1 mL aqueous phase was collected for analysis.

### 2.4. Earthworm Bioaccumulation Experiments of Oxyfluorfen

To evaluate the bioavailability of oxyfluorfen in earthworms under different conditions, three types of soil and aged soil amended with different quantities of BCR (0.5%, 1% and 2% ($w/w$)) were selected to conduct the bioaccumulation experiment. Each treatment was mixed with a rotary shaker. The final spiked concentration of oxyfluorfen in soil was 2.5 mg/kg. Earthworms were cultured in the laboratory for at least 14 days before the experiment. All earthworms were aged 2–3 months and had grown a clitellum, and each earthworm was ca. 300 mg. Twenty earthworms were transferred into each treatment soil mentioned above. The test soils were incubated at $22 \pm 2$ °C for 14 days, and the test vessels were covered with a piece of perforated film. Three replications were conducted for each treatment group.

At the end of the earthworm bioaccumulation experiment, surviving earthworms were rinsed and allowed to purge their gut contents for 24 h on moistened filter paper. After weighing the worms, 2 mL water was added, the mixture was shaken thoroughly for 5 min, and then 10 mL ethyl acetate were added and the mixture was extracted for 30 min. Following extraction, the mixture was left to stand for 10 min and 2 g anhydrous sodium sulfate and 1 g NaCl was added, then extracted using the same method used for soil extraction. The purified solvent was 200 mg Florisil and 150 mg anhydrous sodium sulfate.

### 2.5. Chemical Extraction of Oxyfluorfen in Soil
#### 2.5.1. Tenax Extraction

The operation of Tenax extraction followed previous studies [39]. One gram of each soil sample (unamended and amended soil) was added into separate EPA glass tubes. One mg $HgCl_2$, 40 mL deionized water, and 0.5 g Tenax resins were added, then rotated for 30 min at 60 rpm with a rotary shaker. At periodic intervals (1, 2, 4, 7, 12 24, 48, 96, 168, and 288 h), the Tenax resins were taken out and refreshed, rinsed twice with 10 mL acetone for 15 min, a 10 mL mixture of n-hexane and acetone (1:1, $v/v$) was added to extract, then the

mixtures were dried at 40 °C, dissolved in 2 mL n-hexane and filtered with a 0.22 μm filter for injection.

### 2.5.2. HPCD Extraction

HPCD extraction was conducted with method described by Crampon [40]. One gram of each soil sample was added into separated 50 mL EPA glass tubes, and 25 mL 50 mmol HPCD solution was added. The mixture was shaken at 200 rpm for at least 24 h, then centrifuged 10 min at 7000 rpm. The liquid phase was rotated to dry, dissolved into 2 mL n-hexane, and filtered with a 0.22 μm filter for injection. These concentrations had the code $C_{HPCD}$.

### 2.5.3. Butanol Extraction

Butanol extraction method followed previous studies [41]. One gram of each soil sample was added into separate 50 mL EPA glass tubes, then 15 mL butanol was added and the mixture was shaken for 2 h at 200 rpm, then centrifuged for 30 min at 7000 rpm. The solid phase was extracted using the QuEChERS method, with the concentration coded as $C_{QuEChERS}$. Thus, the concentration of the butanol method was $C_{Butanol} = C_{QuEChERS} - C_B$.

### 2.6. Residue Determination of Oxyfluorfen

The residue of oxyfluorfen in the above samples (including earthworms, soil, Tenax, etc.) was detected using a previously described analysis method [37].

### 2.7. Quantification and Data Analysis

Freundlich isotherm model was used to fit desorption:

$$Q_e = K_f C_e{}^n$$

where $Q_e$ (μg/g) is the amount of oxyfluorfen in the solid phase, $C_e$ (mg/L) is the equilibrium solution concentration, n is an empirical exponent indicative of isotherm nonlinearity, and $K_f{}^{des}$ [(μg/g)/(mg/L)$^n$] is the Freundlich unit capacity coefficient.

Bioaccumulation factor (BCF) was expressed as below:

$$BCF = C_{worm}/C_{soil}$$

where $C_{worm}$ (*g/g*) is the concentration in earthworms (dry weight), and $C_{soil}$ (*g/g*) is the concentration extracted in soil.

A triphasic kinetic model was used to fit data of the consecutive desorption of Tenax:

$$S_t/S_0 = F_r e^{-k1t} + F_{sl} e^{-k2t} + F_{vl} e^{-k3t}$$

where $S_0$ and $S_t$ are the amounts of oxyfluorfen in soil at the start (0) and at time *t* (h) and $F_r$, $F_{sl}$, and $F_{vl}$ are the rapid, slow, and very slow desorbing fractions, respectively.

All statistical data analysis and significance level testing was done with SPSS 25.0 (ANOVA, Tukey's HSD, $p < 0.05$). The model of Freundlich isotherm and triphasic kinetic were performed using Origin 8.5.

Prediction of oxyfluorfen bioavailability was performed using the equilibrium partition theory. Correlation analysis between accumulation in earthworm and extractable concentrations in soil by chemical methods was performed to evaluate the feasibility of each extraction method.

## 3. Result and Discussion

### 3.1. Desorption of the Oxyfluorfen

All desorption isotherms of oxyfluorfen in all soils were fitted with the Freundlich equation ($R^2 > 0.89$, Table 2). The desorption capacity of oxyfluorfen was reduced after addition of the biochar. The desorption coefficient value ($K_f^{des}$) was significantly increased with quantities of BCR ($p < 0.05$). The $K_f^{des}$ of the DBS was increased progressively from 119 $(\mu g/g)/(mg/L)^n$ in the fresh, unamended soil to 439, 516, and 920 $(\mu g/g)/(mg/L)^n$ in 0.5%, 1%, and 2% BCR amended soil, respectively (Table 2). The same trend was also observed in HNS and HBS. For HNS, the $K_f^{des}$ values were 64 in pure soil and 176, 374, and 788 $(\mu g/g)/(mg/L)^n$, respectively, for soil amended with 0.5%, 1%, and 2% BCR. The $K_f^{des}$ values were 85 $(\mu g/g)/(mg/L)^n$ in pure soil, and 225, 393, and 734 $(\mu g/g)/(mg/L)^n$ in soil amended with 0.5%, 1%, and 2% BCR, respectively, for HBS. These data were consistent with the results of published studies: the desorption capacities of the BCR-amended soil were lower than the unamended soil, which may be caused by irreversible sorption of pesticides onto the micro-pores of the biochar [33,34]. The order for desorption capacities was DBS > HBS > HNS, corresponding to the content of organic carbon.

**Table 2.** Parameters values of oxyfluorfen desorption in unamended and biochar-amended soils fitted with Freundlich isotherm model. Data are expressed as the mean values $\pm$ SE (standard error).

| Treatment | Fresh | | | 1 Month | | | 3 Months | | | 6 Months | | |
|---|---|---|---|---|---|---|---|---|---|---|---|---|
| | $K_f^{des}$ | 1/n | $R^2$ | $K_f^{des}$ | 1/n | $R^2$ | $K_f^{des}$ | 1/n | $R^2$ | $K_f^{des}$ | 1/n | $R^2$ |
| DBS | 119 ± 5.2 | 0.67 | 0.89 | 119 ± 4.3 | 0.67 | 0.96 | 120 ± 4.9 | 0.68 | 0.93 | 120 ± 5.4 | 0.66 | 0.93 |
| DBS + 0.5% BCR | 439 ± 9.2 | 0.64 | 0.92 | 398 ± 4.7 | 0.65 | 0.99 | 377 ± 4.3 | 0.64 | 0.95 | 353 ± 4.7 | 0.65 | 0.96 |
| DBS + 1% BCR | 516 ± 7.9 | 0.55 | 0.94 | 499 ± 7.3 | 0.58 | 0.94 | 433 ± 4.9 | 0.61 | 0.96 | 372 ± 6.2 | 0.62 | 0.91 |
| DBS + 2% BCR | 920 ± 8.3 | 0.52 | 0.96 | 868 ± 6.5 | 0.56 | 0.92 | 772 ± 6.9 | 0.57 | 0.99 | 701 ± 7.1 | 0.55 | 0.99 |
| HNS | 64 ± 4.2 | 0.64 | 0.94 | 65 ± 5.2 | 0.62 | 0.97 | 63 ± 7.5 | 0.58 | 0.94 | 65 ± 9.2 | 0.55 | 0.94 |
| HNS + 0.5% BCR | 176 ± 6.5 | 0.64 | 0.95 | 170 ± 7.4 | 0.64 | 0.95 | 154 ± 6.3 | 0.59 | 0.92 | 144 ± 2.4 | 0.58 | 0.99 |
| HNS + 1% BCR | 374 ± 8.7 | 0.62 | 0.97 | 306 ± 8.2 | 0.59 | 0.94 | 245 ± 7.3 | 0.60 | 0.94 | 231 ± 8.8 | 0.54 | 0.97 |
| HNS + 2% BCR | 788 ± 7.4 | 0.59 | 0.98 | 711 ± 4.7 | 0.61 | 0.96 | 641 ± 9.3 | 0.62 | 0.95 | 611 ± 10.3 | 0.52 | 0.94 |
| HBS | 85 ± 4.8 | 0.68 | 0.97 | 83 ± 7.2 | 0.64 | 0.92 | 85 ± 6.3 | 0.61 | 0.92 | 84 ± 6.1 | 0.58 | 0.96 |
| HBS + 0.5% BCR | 225 ± 9.3 | 0.64 | 0.94 | 219 ± 7.5 | 0.62 | 0.89 | 201 ± 9.7 | 0.56 | 0.99 | 197 ± 8.6 | 0.52 | 0.98 |
| HBS + 1% BCR | 393 ± 6.4 | 0.62 | 0.98 | 367 ± 8.6 | 0.58 | 0.94 | 358 ± 11.4 | 0.57 | 0.95 | 298 ± 9.3 | 0.54 | 0.97 |
| HBS + 2% BCR | 734 ± 8.9 | 0.66 | 0.94 | 687 ± 7.5 | 0.56 | 0.91 | 579 ± 9.2 | 0.55 | 0.92 | 501 ± 12.4 | 0.56 | 0.89 |

The desorption capacity of oxyfluorfen increased with aging time in all unamended and BCR-amended soils, which was similar to a previous study [42]. The desorption coefficient value ($K_f^{des}$) of oxyfluorfen decreased from 439 $(\mu g/g)/(mg/L)^n$ in the unaged treatment to 398, 377, and 353 $(\mu g/g)/(mg/L)^n$ after BCR was aged 1, 3, and 6 months, respectively, in 0.5% biochar amended soil. Similar trends were observed in soil amended with 1% and 2% BCR, where the $K_f^{des}$ value ranged from 516 and 920 $(\mu g/g)/(mg/L)^n$ in the unaged treatments and from 372 and 701 $(\mu g/g)/(mg/L)^n$ after 6 months of aging, respectively (Table 2). Similar changes were observed in the other two types of soil: the $K_f^{des}$ values of HBS and HNS in the 6-month treatment were decreased by 6.0 and 9.4 times, respectively, compared with the unaged soil. The results indicated that aged biochar was still effective at desorption. The reason for the increased desorption with aging time might be the interaction of soil minerals and biochar [43]. Additionally, the pesticide was replaced with dissolved organic carbon (DOC) due to the aging treatment in the soil [44], resulting in fewer visible pores after aging, which was also supported by the SEM images in our study (Figure 1).

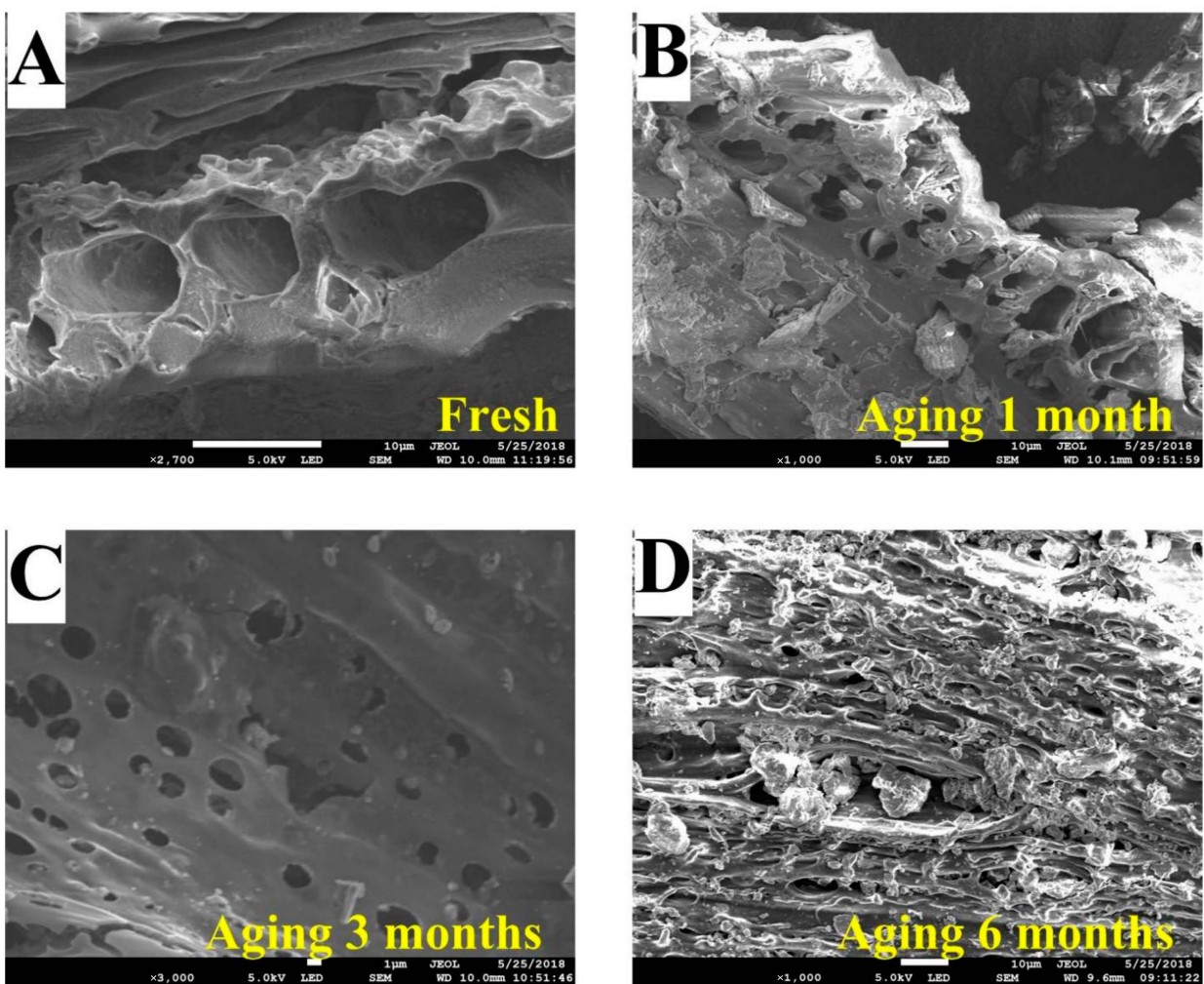

**Figure 1.** Scanning electronic microscopy (SEM) images of fresh biochar (**A**) and biochar after aging 1 month (**B**), 3 months (**C**), and 6 months (**D**) from HNS.

*3.2. Bioaccumulation Experiments*

During the whole observation period, there was no mortality or abnormal behavior in the earthworms in all treatment groups. However, significant differences of BCF were observed among the different types of soil ($F_{2,135}$ = 116.077, $p$ < 0.001). The BCF in the earthworms from DBS was lowest (0.8) compared to others, with the highest found in HNS (1.7); the order of the BCF was HNS > HBS > DBS (Figure 2). In all types of soil, the BCF of the earthworms was decreased in soil amended with BCR (0.10–1.56) compared with unamended soil (0.80–1.7). The bioavailability of the organic contaminants in the soil was reduced after amendment with biochar, consistent with other studies [45,46]. Due to the high adsorption capacity of biochar [47], contamination in pore water decreased after the biochar addition, which caused a lower concentration to accumulate in the living organisms [48], resulting in lower risk. Furthermore, the BCF was reduced with increased biochar application quantities. Moreover, the BCF of the earthworms increased with aging time, from 0.10 to 0.73 in DBS, indicating that the bioavailability was notably influenced by the aging period. However, for HNS, there was no significant difference between unaged soil and soil aged for 1 month ($F_{1,14}$ = 3.937, $p$ = 0.067); with additional aging time, the difference became significant at 6 months ($F_{1,14}$ = 5.155, $p$ = 0.039). This result indicated that oxyfluorfen in soil was released into the pore water after the aging process due to the lower adsorption capacity and higher desorption of the aged biochar [42], consistent with the $Kf^{\text{des}}$ values. The released oxyfluorfen in the soil finally entered the earthworms, causing increased BCF. However, even after aging 6 months, the BCF in BCR-amended soil

was still lower than in unamended soil, indicating that BCR could be a practical method to amend soil.

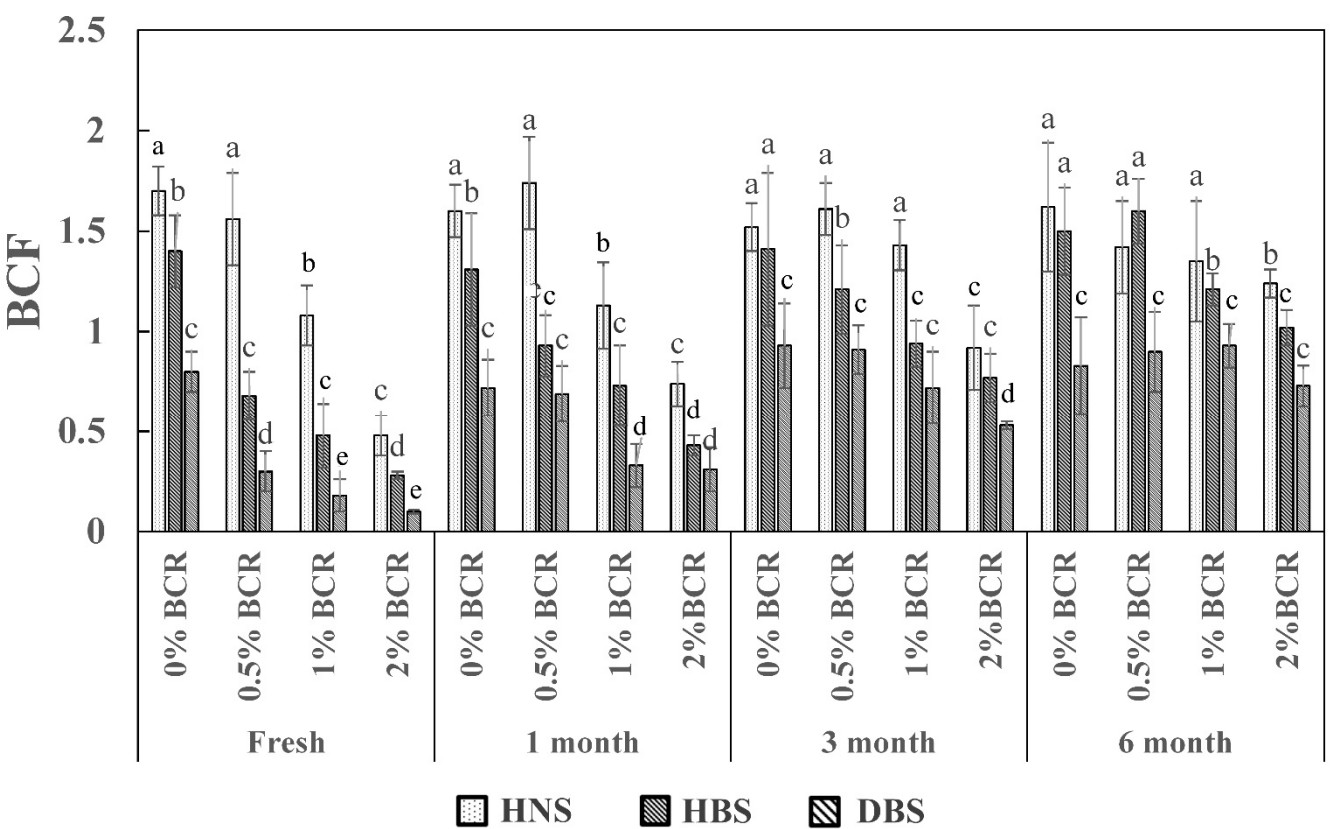

**Figure 2.** Bioaccumulation factor (BCF) of earthworms in three types of soils with biochar (BCR) after aging treatment. Data are expressed as the mean values ± SE (standard error). Different letters indicate significant differences among the treatments (two-way ANOVA, Tukey's HSD, $p < 0.05$).

### 3.3. Desorption of Consecutive Tenax Extractions

At least 200 h are needed to predict the rapid-desorbing fraction by consecutive Tenax extractions [28]; therefore, simplifying the technique is necessary. In our study, rapid desorption of oxyfluorfen was observed in the first 7 h, resulting in 87% residues of oxyfluorfen. Then desorption of oxyfluorfen slowed down. After 96 h of desorption, 89% of the residues were detected in the soil. At the end of desorption, the residue concentration of the oxyfluorfen in the soil was at 92%. The desorption of consecutive Tenax extractions was fitted to a three-compartment model. The rate constants for rapid, slow, and very slow desorption were about 1/10, 1/100, and 1/10,000 $h^{-1}$, respectively (Table 3). Sequential extractions are time-consuming and tedious. A few studies demonstrated that a 6 h extracted fraction from Tenax showed high correlations with the rapid-desorbing fraction [49–51] in all soils. Therefore, a 6 h single-point Tenax extraction was introduced as an alternative. In our study, statistical analysis demonstrated that the 6 h extracted concentration was highly correlated with the concentration of the rapid-desorbing fraction ($R^2 = 0.83$). The linear regression was well fitted, and 0.86 times that of the rapid-desorbing fraction. These results were consistent with other studies, e.g., Cornelissen et al. [49], who found that the 6 h extracted fraction was about 0.5 times that of the rapid-desorbing fraction, which implied that 6 h Tenax could be regarded as an efficient technique to predict bioavailability in soil [52].

**Table 3.** Rate constants and fractions for the rapid, slow, and very-slow desorption of oxyfluorfen in unamended and biochar-amended soil samples predicted by consecutive Tenax extractions.

| Samples, Treatment | Rate Constants and Fractions | | | | | |
|---|---|---|---|---|---|---|
| | $F_r$ | $k_r$ (h$^{-1}$) | $F_{sl}$ | $k_{sl}$ (h$^{-1}$) | $F_{vl}$ | $k_{vl}$ (h$^{-1}$) |
| Fresh | 0.31 | 0.354 | 0.31 | 0.043 | 0.38 | 0.0003 |
| 1 month | 0.31 | 0.453 | 0.25 | 0.032 | 0.44 | 0.0005 |
| 3 months | 0.28 | 0.543 | 0.22 | 0.046 | 0.50 | 0.0004 |
| 6 months | 0.27 | 0.453 | 0.20 | 0.047 | 0.53 | 0.0003 |

$F_r$, $F_{sl}$, and $F_{vl}$ are the rapid, slow, and very slow desorbing fractions, respectively. $k_r$, $k_{sl}$, and $k_{vl}$ (h$^{-1}$) are the first-order rate constants for rapid, slow, and very slow desorption, respectively.

### 3.4. Comparing Chemical Extraction Methods to Predict Bioavailability

Several methods have been selected to estimate bioavailability, such as HPCD, butanol extraction, and Tenax extractions, because sensitivity, cost, and practicability must be considered when choosing an efficient chemical extraction method. Highly significant linear correlations were obtained only for one-point Tenax extraction (6 h). As shown in Figure 3, the concentrations of oxyfluorfen in the earthworms were highly linearly correlated with the concentrations in the soils extracted using Tenax methods ($R^2$ = 0.8604). For other chemical methods, regression coefficients were lower than 0.80 (Figure 3). For example, with QuEChERS, the oxyfluorfen was not only extracted from dissolved concentration in the soil but also extracted by sorption in the soil, which could not be accessed by the earthworms. Therefore, this study suggests that Tenax extractions were more reliable than HPCD and butanol extractions for the bioavailability assessment of oxyfluorfen. Different chemical methods show different extraction mechanisms and thus result in different extraction efficiencies [28], and Tenax extraction was more sensitive in predicting bioavailability due to being more highly linearly correlated with accumulation in the earthworms.

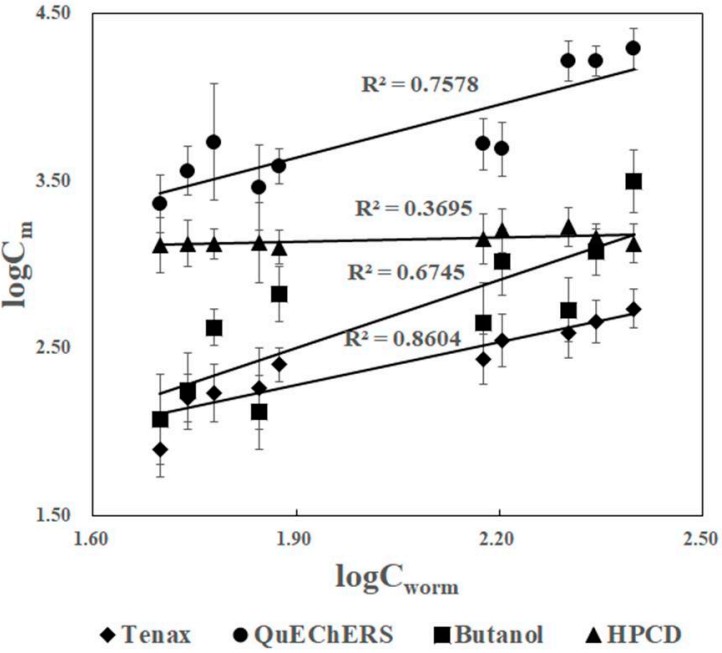

**Figure 3.** Correlation of oxyfluorfen accumulation in earthworms and extracted from soil by different chemical extraction methods. Data are expressed as the mean values ± SE (standard error).

### 3.5. Application to Assess Bioavailability

The one-point Tenax method could be used to estimate the bioavailability of oxyfluorfen. Thus, whether Tenax could be used in aging soil and BCR-amended soil was also investigated. The relationship between Tenax concentration and earthworm concentration

fit linear regression models (Figure 4, $R^2$ = 0.8509) for fresh BCR-amended soil. For chemical extraction, the concentration of oxyfluorfen slightly increased with increased aging periods. A highly significant linear correlation was found between the concentrations extracted from the earthworms and the soils aged for different times (1, 3, and 6 months). However, the regression coefficient $R^2$ decreased with aging time, from 0.8762 to 0.8013, indicating that the bioavailability was notably influenced by the aging period with chemical extraction. The results showed that with increased aging time, the rate of slow desorbing was decreased, but there was no significant change in rapid desorbing (Table 3). The Tenax method mainly involves rapid desorbing, which was not significantly changed, and thus Tenax could predict the bioavailability of oxyfluorfen in the soil even after amending with aged BCR.

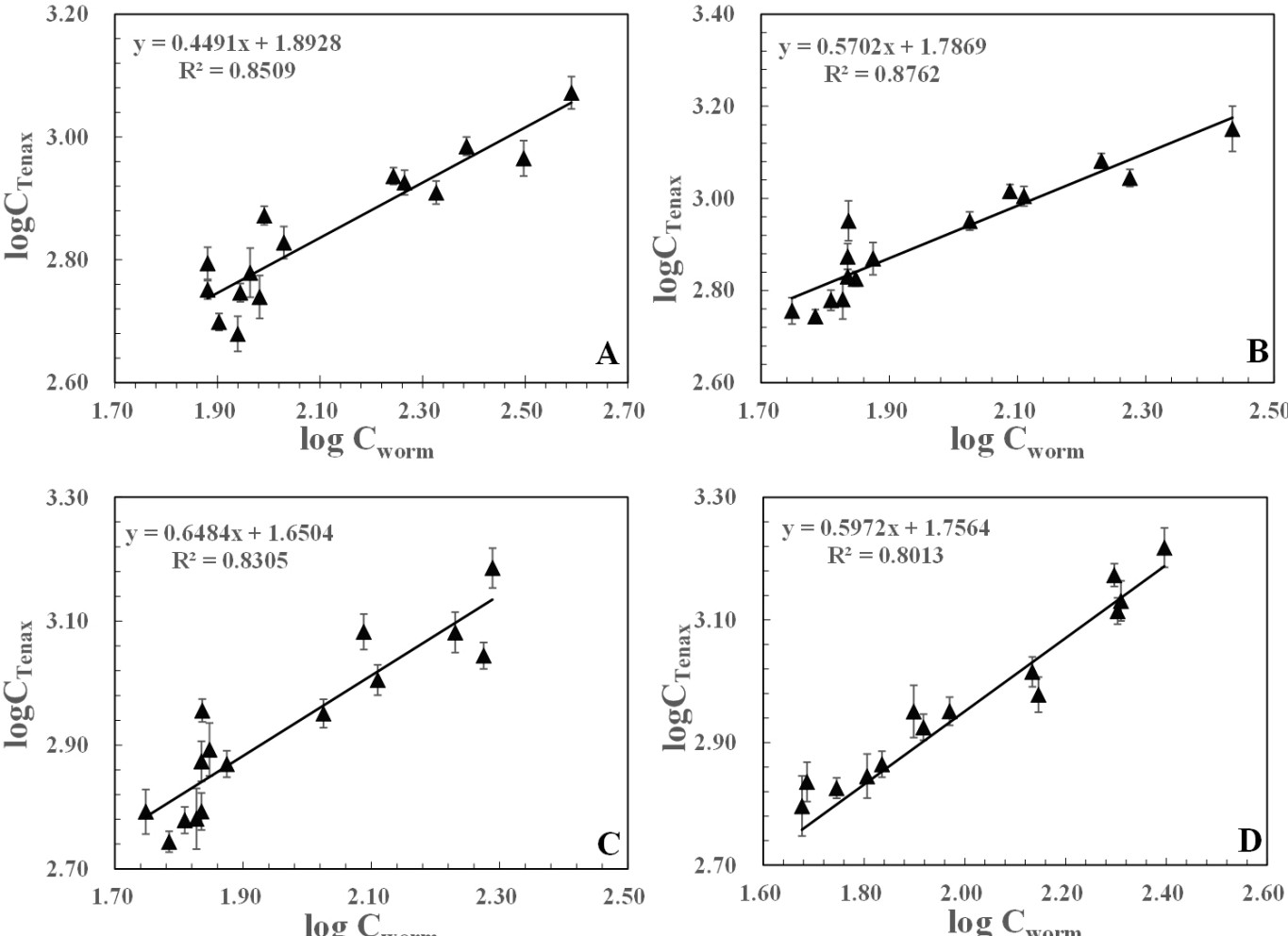

**Figure 4.** Correlation of oxyfluorfen accumulation in earthworms and that extracted from soil ((**A–D**): unaged soil and soil aged 1, 3, and 6 months, respectively) by Tenax methods. Data are expressed as the mean values ± SE (standard error).

## 4. Conclusions

After the addition of BCR to three types of soil, desorption capacity increased and the bioavailability of oxyfluorfen to the earthworms decreased. These changes varied among different types of soil (the order of change was HNS, HBS, and DBS). Increasing the amendment rate of BCR resulted in the desorption coefficient values increasing from 64–119 to 176–920 $(\mu g/g)/(mg/L)^n$, and the BCF of oxyfluorfen in the earthworms was decreased from 0.80–1.7 to 0.10–1.56. After aging, the desorption capacity and bioavailability were both increased. Three kinds of chemical extraction (Tenax, HPCD, QuEChERS, and butanol

extraction) were used to predict bioavailability, and the results indicated that the 6 h Tenax extraction method was highly correlated with the earthworms' uptake of oxyfluorfen ($R^2 = 0.8604$, $p < 0.05$) compared to the other methods. Moreover, the Tenax method was highly linearly correlated with the uptake in the earthworms in all unamended soils and the BCR-amended soil, even after aging ($R^2 > 0.80$). Consequently, BCR could be a practical method to reduce soil contamination, even after 6 months of aging, and the Tenax method could be a sensitive and feasible tool to assess the bioavailability of oxyfluorfen in soil.

**Author Contributions:** Conceptualization, C.W.; methodology, Y.Z. (Yanning Zhang), C.W. and L.M.; validation, L.M., L.Z. (Lan Zhang) and L.Z. (Lizhen Zhu); investigation, Y.Z. (Yanning Zhang) and C.W.; resources, H.J.; data curation, C.W., Y.Z. (Yanning Zhang), L.M. and L.Z. (Lan Zhang); writing—original draft preparation, C.W., Y.Z. (Yanning Zhang), L.Z. (Lan Zhang) and L.Z. (Yanning Zhang); writing—review and editing, Y.Z. (Yongquan Zheng), H.J. and X.L.; visualization, Y.Z. (Yongquan Zheng); supervision, X.L. and Y.Z. (Yongquan Zheng); project administration, X.L. and Y.Z. (Yongquan Zheng); funding acquisition, X.L. All authors have read and agreed to the published version of the manuscript.

**Funding:** This research was funded by National Natural Science Foundation of China grant number 32072466 and Sinograin II project of the Norwegian Ministry of Foreign Affairs through the Royal Norwegian Embassy in Beijing grant number CHN 2152, 17/0019.

**Institutional Review Board Statement:** Not applicable.

**Informed Consent Statement:** Not applicable.

**Data Availability Statement:** The data presented in this study are available in Table and Figure of article.

**Acknowledgments:** This work was supported by the National Natural Science Foundation of China (32072466) and the Sinograin II project [CHN 2152, 17/0019] of the Norwegian Ministry of Foreign Affairs through the Royal Norwegian Embassy in Beijing.

**Conflicts of Interest:** The authors declare that they have no known competing financial interests or personal relationships that could have influenced the work reported in this paper.

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
