# Peer review of "Efficiency of Four Extraction Methods to Assess the Bioavailability of Oxyfluorfen to Earthworms in Soil Amended with Fresh and Aged Biochar"

_agriculture, doi:10.3390/agriculture12060765_

Round 1

Reviewer 1 Report

Manuscript ID: agriculture-1741137

The authors improved the manuscript, changing awkward sentences and clarifying others. However, the statistical analysis and results are still unclear and lack correct reports.

In lines 168-170, they mention ‘Regression lines were compared by analysis of covariance to determine equality of slopes and intercept’. It doesn't make any sense. ANCOVA uses a similar approach as ANOVA but with a variable that is related to the other – like categorical and continuous as independent variables for the same dependent (e.g., continuous). What the authors did, I assume, is expressed in the captions of the figures. 

In figure 2 (lines 420-422) they used ANOVA followed by the Tukey test to compare the variations of BCF according to soil and BCR percentual (two-away ANOVA? – not mentioned either). And please, avoid the use of acronyms in the caption, we may need to look back to understand what BCF and BCR mean. In figure 3, it was performed a correlation analysis (lines 423-425), similar to figure 4 (427-429).

The regression mentioned in lines 168-170 might be related to table 2 but needs to be precisely indicated.

I strongly recommend that the authors mention these analyses correctly in lines 168-170, indicating for each the independent and dependent variables involved. Additionally, in the results, I expect the mention of F-values, P-values, and degrees of freedom related to each analysis to be clear to readers how the analysis was correctly done. P-value per se (p<0.05) does not bring confidence to the reports.

Author Response

The authors improved the manuscript, changing awkward sentences and clarifying others. However, the statistical analysis and results are still unclear and lack correct reports.

In lines 168-170, they mention ‘Regression lines were compared by analysis of covariance to determine equality of slopes and intercept’. It doesn't make any sense. ANCOVA uses a similar approach as ANOVA but with a variable that is related to the other – like categorical and continuous as independent variables for the same dependent (e.g., continuous). What the authors did, I assume, is expressed in the captions of the figures. 

A: Thanks for your reminder. This part has been revised to “Correlation analysis between accumulation in earthworm and extractable concentrations in soil by chemical methods was performed to evaluate the feasibility of each extraction method.”

In figure 2 (lines 420-422) they used ANOVA followed by the Tukey test to compare the variations of BCF according to soil and BCR percentual (two-away ANOVA? – not mentioned either). And please, avoid the use of acronyms in the caption, we may need to look back to understand what BCF and BCR mean. In figure 3, it was performed a correlation analysis (lines 423-425), similar to figure 4 (427-429).

A: Thanks for your kindly reminder. The figure captions have been revised

“Figure 2. Bioaccumulation factor (BCF) of earthworms in three types of soils with biochar (BCR) after aging treatment. Data are expressed as the mean values ± SE (standard error). Different letters indicate significant differences among the treatments (two-way ANOVA, Tukey's HSD, P<0.05).

Figure 3. Correlation of oxyfluorfen accumulation in earthworms and extracted from soil by dif-ferent chemical extraction methods. Data are expressed as the mean values ± SE (standard error).

Figure 4. Correlation of oxyfluorfen accumulation in earthworms and extracted from soil (A to D-unaged soil and soil aged 1, 3, and 6 months) by Tenax methods. Data are expressed as the mean values ± SE (standard error).”

The regression mentioned in lines 168-170 might be related to table 2 but needs to be precisely indicated.

A: Due to model of Freundlich isotherm was performed using Origin 8.5. Correlation analysis between accumulation in earthworm and extractable concentrations in soil by chemical methods was performed to evaluate the feasibility of each extraction method which also mentioned in MS.

“All statistical data analysis and significance level testing was done with SPSS 25.0 (ANOVA, Tukey's HSD, P<0.05). The model of Freundlich isotherm and triphasic kinetic were performed using Origin 8.5.

Prediction of oxyfluorfen bioavailability was done using the equilibrium partition theory. Correlation analysis between accumulation in earthworm and extractable concentrations in soil by chemical methods was performed to evaluate the feasibility of each extraction method.

 “Table 2. Parameters values of oxyfluorfen desorption in unamended and biochar-amended soils fitted with Freundlich isotherm model. Data are expressed as the mean values ± SE (standard error).”

I strongly recommend that the authors mention these analyses correctly in lines 168-170, indicating for each the independent and dependent variables involved. Additionally, in the results, I expect the mention of F-values, P-values, and degrees of freedom related to each analysis to be clear to readers how the analysis was correctly done. P-value per se (p<0.05) does not bring confidence to the reports.

A: Thanks for your kindly suggestion. The description has been added in MS. And the F-values, P-values, and degrees of freedom related to each analysis have been added in results.

“Prediction of oxyfluorfen bioavailability was done using the equilibrium partition theory. Correlation analysis between accumulation in earthworm and extractable concentrations in soil by chemical methods was performed to evaluate the feasibility of each extraction method.”

“During the whole observation period, there was no mortality or abnormal behavior of earthworms in all treatment groups. However, significant differences of BCF were observed among different types of soil (F2,135=116.077, p<0.001). The BCF in earthworms from DBS was lowest (0.8) compared to others, with the highest found in HNS (1.7); the order of BCF was HNS>HBS>DBS (Fig. 2). In all types of soil, the BCF of earthworms was decreased in soil amended with BCR (0.10-1.56) compared with unamended soil (0.80-1.7). Bioavailability of organic contaminants in soil was reduced after amendment with biochar, consistent with other studies [45,46]. Due to the high adsorption capacity of biochar [48], contamination in pore water decreased after biochar addition, which caused a lower concentration to accumulate in living organisms [49] resulting in lower risk. Furthermore, the BCF was reduced with increased biochar application quantities. Moreover, the BCF of earthworms increased with aging time, from 0.10 to 0.73 in DBS, indicating that the bioavailability was notably influenced by the aging period. However, for HNS, there was no significant difference between unaged soil and soil aged for 1 month (F1,14=3.937, p=0.067); with additional aging time the difference became significant at 6 months (F1,14=5.155, p=0.039).”

Reviewer 2 Report

Comments

1. The writing is not up to standard. Comprehensive English language and style editing is necessary for the entire manuscript.

2. Put references at line 59

3. There are double section of 2.4, please correct it…

4. Put references at methodology section- Chemical extraction of oxyfluorfen in soil

5. At line 265-267….The results showed that with increased aging time, the 265

rate of slow desorbing was decreased but there was no significant change in rapid-desorbing…Please explan why..Please also support your results with another references…

6. At line 254, why Tenax extraction was more sensitive in predicting bioavailability? Please explain.

7. At line 220….Even after aging 6 months, the BCF in BCR amended soil was still lower than in 220 unamended soil…Please explain?

8. At line 248, please explain why regression coefficients were lower than 0.80..

Author Response

  1. The writing is not up to standard. Comprehensive English language and style editing is necessary for the entire manuscript.

 A: Thanks for your kindly comments, the MS has been polished by professional institution and Yanli Man. The certificates as below

  1. Put references at line 59

 A: Thanks for your suggestion. This sentence has been revised to ”As Li et al. reported, bioaccumulation of acetochlor in plants increased when biochar aged for 20 days was used, causing high potential exposure risks to the environment [27]”

  1. There are double section of 2.4, please correct it…

A: Sorry for that and thanks for your kindly reminder. The number of sections has been revised.

  1. Put references at methodology section- Chemical extraction of oxyfluorfen in soil

A: The related references have been added, Thanks for your kindly suggestion.

“40. Jia, F., Liao, C., Xue, J., Taylor, A., Gan, J., Comparing different methods for assessing contaminant bioavailability during sediment remediation. Sci. Total Environ. 2016. 573, 270-277.

  1. Crampon M, Bodilis J, Le Derf F, Portet-Koltalo F. Alternative techniques to HPCD to evaluate the bioaccessible fraction of soil associated PAHs and correlation to biodegradation efficiency. J Hazard Mater. 2016. 314:220–229
  2. Kelsey JW, Kottler BD, Alexander M. Selective chemical extractants to predict bioavailability of soil-aged organic chemicals. Environ Sci Technol. 1997. 31:214–217”

5. At line 265-267….The results showed that with increased aging time, the 265

rate of slow desorbing was decreased but there was no significant change in rapid-desorbing…Please explan why..Please also support your results with another references…

A: As the result in Table 3. The with increased aging time, the rate of slow desorbing was decreased but there was no significant change in rapid-desorbing which could support our results of bioavailability (Tenax could predict the bioavailability of oxyfluorfen in soil even after amending with aged BCR). As best our knowledge, there was no other reference related to our results.

In our study, due to rapid desorption of consecutive Tenax extractions could roughly predict the concentration in soil, and 6h single-point Tenax method could regard as consecutive Tenax, which are given in Section 3.3. Meanwhile, with increased aging time, the rate of slow desorbing was decreased but there was no significant change in rapid-desorbing (Table 3).

6..  At line 254, why Tenax extraction was more sensitive in predicting bioavailability? Please explain

A: Tenax extraction was more sensitive in predicting bioavailability due to higher linearly correlated with accumulated in earthworm. As Figure 3 showed, the regression coefficient of Tenax 0.8604 was higher than other chemical extraction methods (0.7578, 0.3695, 0.6745).

  1. At line 220….Even after aging 6 months, the BCF in BCR amended soil was still lower than in 220 unamended soil…Please explain?

A: Although oxyfluorfen in soil was released into the pore water after aged, the BCR showed high sorption capacity, less oxyfluorfen in pore-water of soil which could accumulate into earthworm than soil without BCR. Thus, BCF in BCR was still lower in unamended soil. BCR could be practical material during remediation of soil.

  1. At line 248, please explain why regression coefficients were lower than 0.80.

A: As Figure 3 showed, the regression coefficient of Tenax 0.8604 was higher than other chemical extraction methods (0.7578, 0.3695, 0.6745).

Reviewer 3 Report

The manuscript entailed- “Efficiency of four extraction methods to assess the bioavailability of oxyfluorfen to earthworms in soil amended with fresh and aged biochar” is critically re-reviewed and found to be a very interesting study. The manuscript is significantly modified.

Author Response

Thanks for your kindly suggestion, it is very help for our furture study.

Round 2

Reviewer 2 Report

The manuscript is well improved